# Comparison of Efficacy and Safety of Brigatinib in First-Line Treatments for Patients with Anaplastic Lymphoma Kinase-Positive Non-Small-Cell Lung Cancer: A Systematic Review and Indirect Treatment Comparison

**DOI:** 10.3390/jcm11112963

**Published:** 2022-05-24

**Authors:** Yongfeng Yu, Fanfan Zhu, Wenxin Zhang, Shun Lu

**Affiliations:** 1Shanghai Lung Cancer Center, Shanghai Chest Hospital, Shanghai Jiao Tong University, Shanghai 200052, China; yuyongfeng212@sjtu.edu.cn; 2Takeda (China) International Trading Co., Ltd., Shanghai 200124, China; fanfan.zhu@takeda.com (F.Z.); wendy.zhang@takeda.com (W.Z.)

**Keywords:** systematic review, indirect treatment comparison, tyrosine kinase inhibitors, *ALK*-positive non-small-cell lung cancer

## Abstract

(1) Background: The relative efficacy and safety of brigatinib compared with other next-generation anaplastic lymphoma kinase (ALK) inhibitors remains unclear, as first-line head-to-head trials have not been conducted. (2) Methods: Electronic databases were systematically searched for eligible randomized controlled trials (RCT) from January 2010 to October 2021. Outcomes evaluated by indirect treatment comparison (ITC) included progression-free survival (PFS), overall survival (OS), objective response rate (ORR), and safety. (3) Results: Nine RCTs with 2484 patients assessing crizotinib, ceritinib, alectinib, brigatinib, ensartinib, and lorlatinib were included. In intent-to-treat (ITT) patients, brigatinib significantly prolonged blinded independent review committee-assessed PFS compared with crizotinib (HR: 0.48, 95% CI: 0.35 to 0.66) and ceritinib (HR: 0.38, 95% CI: 0.23, 0.60) and had a comparable PFS with other 2nd-generation ALK inhibitors. Subgroup analyses of patients with baseline brain metastases and Asian patients yielded results similar to the base case. Brigatinib significantly reduced the risk of death compared with crizotinib (HR: 0.50, 95% CI: 0.28, 0.87) after adjusting for treatment crossover in the crizotinib arm. No significant differences were observed in OS between brigatinib and other next-generation ALK inhibitors. Brigatinib had significantly superior effects in ORR and intracranial ORR compared to crizotinib. The incidence of grade ≥3 AEs was similar between brigatinib and other next-generation ALK inhibitors (except for alectinib), while brigatinib could significantly delay the time to worsening in the European Organization for Research and Treatment of Cancer Quality of Life Questionnaire-Core 30 (EORTC QLQ-C30) global health status (GHS)/quality of life (QoL) vs. crizotinib (HR: 0.69, 95% CI: 0.49, 0.98). (4) Conclusions: Brigatinib had longer PFS compared to crizotinib and ceritinib and had comparable efficacy and safety profile with other 2nd-generation ALK inhibitors in first-line treatments for patients with *ALK-*positive non-small-cell lung cancer.

## 1. Introduction

Lung cancer is the most common cancer and the leading cause of cancer mortality worldwide. In 2018, an estimated 2,093,876 new cases and 1,761,007 deaths related to lung cancer were observed all over the world, representing 11.6% of all new cancer diagnoses and 18.4% of all cancer-related deaths [1]. Non-small-cell lung cancer (NSCLC) is the most common form of lung cancer, accounting for approximately 85% of cases [2]. Anaplastic lymphoma kinase (*ALK*) was validated as a well-established molecular and therapeutic target in several *ALK-*rearranged malignancies, including NSCLC. The incidence of *ALK* rearrangement in Chinese NSCLC patients was reported to be 5.6% [3]. Patients with *ALK*-positive NSCLC feature a more advanced stage and have a higher risk of developing brain metastases than patients with other NSCLC sub-types, thus having a lower quality of life and a poorer prognosis [4].

The introduction of tyrosine kinase inhibitors (TKIs) has revolutionized the treatment of advanced NSCLC with *ALK* rearrangements. Crizotinib is the first-generation ALK inhibitor for the first-line treatment for advanced *ALK*-positive NSCLC patients. However, drug resistance develops after crizotinib’s initial benefits, particularly in the central nervous system (CNS) [5,6]. Several next-generation ALK inhibitors, including ceritinib, alectinib, brigatinib, ensartinib, and lorlatinib have been developed. Many large-scale phase III randomized controlled trials (RCTs) [7,8,9,10,11,12,13,14,15] have shown that next-generation ALK inhibitors have better clinical efficacy, CNS penetration, and safety profiles than the first-generation one.

Brigatinib, a newly developed next-generation ALK inhibitor that was launched in China recently also demonstrated better clinical efficacy than first-generation ALK inhibitors in first-line treatment for patients with advanced *ALK*-positive NSCLC. According to the findings from a multinational phase III study (i.e., ALTA-1L), brigatinib significantly prolonged blinded independent review committee (BIRC)-assessed progression-free survival (PFS) compared to crizotinib (HR: 0.48, 95% CI: 0.35 to 0.66) [15]. Brigatinib demonstrated significantly longer intracranial PFS compared to crizotinib in patients with any brain metastases at baseline by BIRC assessment (HR: 0.29, 95% CI: 0.17–0.51) [15]. In addition, its tolerability profile was manageable, and no new safety concerns were identified [15]. While multiple trials have proven superiority versus crizotinib, head-to-head comparisons of brigatinib and other next-generation ALK inhibitors have not been conducted. Indirect treatment comparison (ITC) [16] is a method of deriving a comparative estimate between treatments that are not directly compared in head-to-head trials.

We adopted the ITC method to evaluate the relative efficacy and safety of brigatinib compared to other ALK inhibitors for the first-line treatment of patients with *ALK*-positive NSCLC. The findings of this study may help clinicians to develop individualized treatments with ALK inhibitors for diverse patients with advanced NSCLC.

## 2. Methods

We followed the International Society for Pharmacoeconomics and Outcomes Research (ISPOR) good practices guideline [17] to conduct a systematic review and ITC and assessed the completeness of this report using the Preferred Reporting Items for Systematic Review and Meta-Analysis (PRISMA) reporting checklist [18].

### 2.1. Search Strategy and Selection Criteria

Electronic databases were systematically searched from January 2010 to October 2021. Studies were considered if they matched the following inclusion criteria: (1) ALK-inhibitor-naïve *ALK*-positive NSCLC patients; (2) either ALK inhibitors or chemotherapy were included in the control arms; (3) phase III RCTs with PFS, overall survival (OS), objective response rate (ORR), and safety profile reported. The detailed search strategies are listed in Appendix A. PICOS are listed in Appendix A. We also manually checked reference lists of related review articles and published trials to identify additional studies. Additional conference abstracts and posters were searched to identify the results of clinical trials not yet published in full text from the American Society of Clinical Oncology (ASCO), European Society for Medical Oncology (ESMO), World Conference on Lung Cancer (WCLC), and Chinese Society of Clinical Oncology (CSCO).

### 2.2. Data Extraction and Quality Assessment

Data were extracted by two independent investigators (Sun, R. and Wu, Y.), and discrepancies were resolved by consensus or by involving a third investigator (Qu, S.). The extracted information included the study name, published year, phase, sample size, patient baseline characteristics, treatment characteristics, hazard ratios (HRs) of PFS and OS, ORR, and safety profiles. The qualities of the studies were evaluated using the domains outlined in the *Cochrane Handbook for Systematic Reviews of Interventions* [19]. When more than one article reported the same outcome, the most recent data were selected.

### 2.3. Outcome Measures and Statistical Methods

The parameters in this study were PFS, OS, ORR, and safety outcomes, such as the proportion of patients experiencing adverse events (AEs) of grade 3 or higher, AEs leading to discontinuation, and AEs leading to dose reduction. Survival outcomes, OS and PFS, were reported with hazard ratios (HRs) with 95% confidence intervals (CIs). ORRs and the incidence of AEs were calculated with odds ratios (ORs). When the 95% CI for an indirect comparison contained 0 or 1, the difference was considered not statistically significant.

All statistical analyses were performed by STATA (V.14.0; StataCorp LLC, College Station, TX, USA). Adjusted indirect comparisons between brigatinib and alectinib, ensartinib, and lorlatinib were conducted using crizotinib as the common comparator. The relative benefit of alectinib versus crizotinib was demonstrated by pooled results of ALEX, ALESIA, and J-ALEX and was further discussed by a different dosage of alectinib. We linked crizotinib and ceritinib using chemotherapy. The statistical heterogeneity in the included studies was assessed using a chi-square test (Q test). The relative effects of different ALK inhibitors in Asian patients and patients with baseline brain metastasis were also investigated in subgroup analyses.

## 3. Results

### 3.1. Studies Included in the ITC

In total, 371 records were identified from the search. We included 15 records, after full-text evaluation, that met the inclusion criteria pertaining to nine unique RCTs (ALTA-1L, ALEX, J-ALEX, ALESIA, PROFILE1014, PROFILE1029, ASCEND-4, eXalt3, and CROWN) with 2484 patients.

The PRISMA flow diagram is presented in Appendix A. The study characteristics of the nine RCTs were summarized in Appendix A. The network plot of the included studies is presented in Figure 1.

Three RCTs (PROFILE1029 [20], J-ALEX [8], and ALESIA [9]) had only enrolled participants from Asia, while six other RCTs [5,7,10,11,12,14] recruited participants globally. The median ages of the participants ranged from 48 to 61 years. The sex and ECOG statuses were relatively balanced across the studies. Most studies predominantly enrolled participants who were stage IV (72–97%) at trial entry and had adenocarcinoma (90–99%). Patients were stratified according to baseline brain metastases (present or absent) in all the trials, and intracranial response was reported in ALTA-1L, CROWN, ALEX, ALESIA, and ASCEND-4. In five RCTs [5,7,8,15,20], participants could cross over to receive the alternative treatment after disease progression (Appendix A).

### 3.2. Risk of Bias

The risk of bias is summarized in Appendix A. All studies properly reported randomized sequence generation and were at low risk for selection bias. Open-label studies were considered to be at high risk of bias in performance.

### 3.3. ITC Results of Efficacy Endpoints

#### 3.3.1. Progression-Free Survival

Intent-to-treat (ITT) population

The ITC results suggested that brigatinib significantly prolonged independent review committee-assessed PFS compared to crizotinib (HR: 0.48, 95% CI: 0.35 to 0.66) and ceritinib (HR: 0.38, 95% CI: 0.23, 0.60). The PFS of brigatinib was comparable to ensartinib (HR: 0.94, 95% CI: 0.58, 1.52) and alectinib (HR: 1.21, 95% CI: 0.83, 1.76). Lorlatinib had a longer PFS than brigatinib (HR: 1.71, 95% CI: 1.04, 2.82). As the dose of alectinib (300 mg twice daily) used in J-ALEX was lower than the recommended dose in countries other than Japan, in comparisons between brigatinib and different doses of alectinib, brigatinib was not inferior to either high-dose or low-dose alectinib in PFS (HR: 1.16, 95% CI: 0.77, 1.75 for high-dose alectinib and HR: 1.30, 95% CI: 0.81, 2.08 for low-dose alectinib) (Figure 2a).

Asian Population subgroup

For the Asian population, brigatinib had a significantly longer PFS compared to crizotinib (HR: 0.35, 95% CI: 0.20, 0.59) and ceritinib (HR: 0.22, 95% CI: 0.10, 0.48). The PFS of brigatinib was comparable to alectinib, regardless of dose, and ensartinib, while the PFS of brigatinib was numerically higher than lorlatinib (HR: 0.80, 95% CI: 0.36, 1.77), although the difference was not statistically significant (Figure 2b).

Baseline brain metastases population

For patients with baseline brain metastases, brigatinib had a longer PFS compared to crizotinib (HR: 0.25, 95% CI: 0.14, 0.46) and ceritinib (HR: 0.19, 95% CI: 0.08, 0.45) and a PFS comparable to lorlatinib (HR: 1.25, 95% CI: 0.49, 3.20) (Figure 2c). PFS was comparable between brigatinib and alectinib (HR: 1.07, 95% CI: 0.35, 3.26), and further analysis showed that PFS with brigatinib was numerically higher than alectinib when only global clinical trial data (ALTA-1L and ALEX) were included for indirect comparison (HR: 0.65, 95% CI: 0.29, 1.45) (Appendix A).

#### 3.3.2. Overall Survival

At the data cutoff of the interim analysis for the ALTA-1L, eXalt3, and CROWN studies, the median OS was not reached in either the treatment or comparison group. Treatment crossover was allowed in five trials (ALTA-1L, J-ALEX, PROFILE1014, PROFILE1029, and ASCEND-4) but not in the other four (ALEX, ALESIA, eXalt3, and CROWN). Thus, we used the HRs of OS for brigatinib to compare with the ALK inhibitors in trials that permitted crossover and adjusted crossover HRs of OS for brigatinib to compare with that of other ALK inhibitors in trials that did not permit crossover.

The ITC results suggested that brigatinib significantly reduced the risk of death compared to crizotinib (HR: 0.50, 95% CI: 0.28, 0.87) after adjusting for treatment crossover in the crizotinib arm by the inverse probability of censoring weight (IPCW) approach [15]. The OS of brigatinib was not significantly different from ceritinib (HR: 0.89, 95% CI: 0.47, 1.67), ensartinib (HR: 0.55, 95% CI: 0.25, 1.19), alectinib (HR: 1.05, 95% CI: 0.38, 2.89 for high-dose, HR: 0.79, 95% CI: 0.43,1.43 for low-dose), or lorlatinib (HR: 0.69, 95% CI: 0.31, 1.54) (Figure 3).

#### 3.3.3. Objective Response Rate

The ORR was defined as the objective response at one or more assessments, including confirmed and unconfirmed responses, and was the secondary endpoint in most ALK inhibitor trials. Our analysis used OR as the effect measure for ORR, with an OR greater than 1.0 indicating an improved outcome. The results suggested that brigatinib was associated with a better ORR than crizotinib (OR: 1.73, 95% CI: 1.04, 2.88). There were no significant differences between brigatinib and the other ALK inhibitors (Figure 4a).

Patients with measurable brain metastases at baseline

For patients with measurable brain metastases at baseline, brigatinib had significantly superior effects in intracranial ORR than crizotinib (OR: 11.67, 95% CI: 2.15, 63.27). Little differences were found in intracranial ORR when comparing brigatinib to high-dose alectinib (OR = 1.19, 95% CI: 0.09, 15.59), ensartinib (OR: 1.79, 95% CI: 0.20, 16.02), and lorlatinib (OR = 0.70, 95% CI: 0.07, 7.06) (Figure 4b).

#### 3.3.4. Safety Profile

Grade ≥ 3 AEs

The ITC results for safety profiles suggested that the incidence of grade ≥3 AEs of brigatinib was comparable to ceritinib (OR = 0.41, 95% CI: 0.09, 1.91), ensartinib (OR = 1.45, 95% CI: 0.71, 2.94), and lorlatinib (OR = 0.91, 95% CI: 0.43, 1.89), while crizotinib and alectinib presented a lower incidence rate of grade ≥3 AEs (OR = 1.99, 95% CI: 1.17, 3.41 for crizotinib, OR = 3.72, 95% CI: 1.77, 7.83 for alectinib).

AEs leading to discontinuation

The incidence of AEs leading to the discontinuation of brigatinib was comparable to other ALK inhibitors (except for low-dose alectinib) (OR: 1.51, 95% CI: 0.70, 3.26 for crizotinib, OR: 6.96, 95% CI: 0.83, 58.09 for ceritinib, OR: 2.73, 95% CI: 0.96, 7.72 for alectinib, OR: 1.11, 95% CI: 0.35, 3.50 for ensartinib, OR: 1.97, 95% CI: 0.63, 6.16 for lorlatinib). (Figure 5a).

AEs leading to dose reduction

The incidence of AEs leading to a dose reduction of brigatinib was comparable to ensartinib (OR: 1.88, 95% CI: 0.88, 4.03) and lorlatinib (OR: 1.57, 95% CI: 0.71, 3.47). Alectinib and crizotinib showed significantly lower incidence rates of AEs leading to dose reduction compared to brigatinib (Figure 5b). The incidence rate of AEs leading to dose reduction was greater in the brigatinib group than that in the crizotinib group, possibly due to more stringent protocol-mandated dose modifications for laboratory abnormalities with brigatinib as compared to crizotinib modifications, which followed standard labeling [13]. Of note, the safety profiles of ALK inhibitors are not identical. As a result, the reasons for AEs leading to dose reduction or discontinuation differ among the ALK inhibitors.

Health-related quality of life (HRQoL)

In addition, brigatinib outperformed crizotinib in terms of health-related quality of life (HRQoL), with a significantly longer duration of improvement in global health status (GHS)/quality of life (QoL) (median time to worsening in GHS/QoL for brigatinib was 26.7 months and for crizotinib was 8.3 months, HR = 0.69, 95% CI: 0.49, 0.98, log-rank *p* = 0.047) and significantly delayed the time to the worsening of emotional and social functioning and symptoms of fatigue, nausea and vomiting, appetite loss, and constipation (log-rank *p* < 0.05) [15]. Due to the lack of patient-level data, no further ITC analyses of HRQoL between brigatinib and other ALK inhibitors were performed.

## 4. Discussion

The existence of an *ALK* gene fusion in individuals with NSCLC predicts a therapeutic benefit from ALK inhibitor therapy [21]. Several ALK inhibitors have demonstrated significant benefits in the treatment of *ALK-*positive NSCLC patients in recent years. Despite the fact that crizotinib, a first-generation ALK inhibitor, is more effective than conventional chemotherapies, almost all patients treated with crizotinib eventually develop resistance [22,23], leading to disease progression and the development of CNS metastases. To achieve improved outcomes and to address the challenge of resistance and CNS progression in crizotinib therapy, the second generation of ALK inhibitors, such as ceritinib, alectinib, ensartinib, and brigatinib, as well as the third-generation lorlatinib were developed.

With increasing treatment options available for patients with *ALK*-positive NSCLC, differentiating among these therapies to guide clinical practice is important. We conducted an ITC analysis to compensate for the evidence gap.

The results of our analysis are consistent with several other network meta-analyses in patients with *ALK*-positive NSCLC. A recent ITC analysis of next-generation ALK inhibitors (alectinib, brigatinib, ensartinib, and lorlatinib) showed that all next-generation ALK inhibitors have longer PFS compared to crizotinib. The third-generation ALK inhibitor lorlatinib was associated with a longer PFS compared to other ALK inhibitors. In the baseline brain metastases subgroup, the PFS benefit was not significantly different among the next-generation ALK inhibitors [24]. Another ITC analysis also suggested that there were no significant differences in PFS between brigatinib and alectinib. However, the surface under the cumulative ranking (SUCRA) values revealed that brigatinib ranked the highest by efficacy in the CNS metastasis subgroup [25]. Of note, these studies have not included all the available ALK inhibitors. Moreover, the efficacy of brigatinib compared to other ALK inhibitors had not been evaluated in Asian patients.

Our study included all the ALK inhibitors of the 1st to 3rd generations that have been approved in China and further investigated the efficacy of brigatinib and other ALK inhibitors among Asian patients and baseline brain metastases patients. The results of our analysis showed that in the first-line treatment of patients with *ALK*-positive NSCLC, brigatinib had better efficacy than crizotinib and ceritinib, and there was no significant difference in terms of PFS, OS, and ORR between brigatinib and other next-generation ALK inhibitors.

Subgroup analyses suggested that the Asian population can benefit more from brigatinib in terms of PFS compared to other ALK inhibitors used as first-line treatments, and patients with measurable brain metastases at baseline can achieve better ORR using brigatinib as the first-line treatment. A numerically higher PFS with brigatinib compared with low-dose alectinib and ensartinib was observed in patients with baseline brain metastases, though the difference was not statistically significant. Subgroup analyses also suggested that brigatinib has a PFS benefit comparable with lorlatinib, while an analysis in ITT patients indicated that brigatinib is inferior to lorlatinib in PFS. Furthermore, we have noticed that brigatinib demonstrated longer intracranial PFS versus crizotinib in patients with any brain metastases at baseline (HR: 0.29; 95% CI: 0.17, 0.51) [15]. Given the lack of a similar demonstration for other 2nd-generation ALK inhibitors, no further ITC was conducted. Future studies are expected to supplement this data gap.

It is worth mentioning that the ITC results of PFS between brigatinib and other ALK inhibitors take into account BIRC-assessed results, while the most recent results from the ALEX study only updated the investigator-assessed results [10]. Thus, we also performed an ITC analysis using the interim BIRC-assessed PFS results from ALEX, with the results suggesting that brigatinib may provide more benefit in patients with baseline brain metastases than alectinib (HR: 0.65, 95%CI: 0.29, 1.45) (Appendix A).

The specific adverse events may differ among ALK inhibitors. Under comparable efficacy circumstances, it is crucial for practitioners to choose the most appropriate ALK inhibitors according to the drugs’ safety profiles and individual patient characteristics. We have noticed that brigatinib is the first ALK inhibitor to demonstrate not only significantly improved efficacy but also significantly improved HRQoL compared to crizotinib [26]. An ITC analysis has not been performed on HRQoL due to the lack of patient-level data.

One limitation of our study that merits consideration is that the data on median OS were often immature in the included studies. Besides, treatment crossover was permitted in a few studies. Thus, we used HRs of OS for brigatinib that adjusted for the crossover effect to compare with that of other ALK inhibitors in trials that did not permit crossover.

## 5. Conclusions

Brigatinib was superior to crizotinib and ceritinib in PFS and ORR and had an efficacy and safety profile comparable to other 2nd-generation ALK inhibitors in first-line treatments for patients with *ALK*-positive NSCLC. Under conditions of greater heterogeneity in patients and treatment settings, we predict more head-to-head studies of ALK inhibitors and real-world evidence to supplement trial evidence.

## Figures and Tables

**Figure 1 jcm-11-02963-f001:**
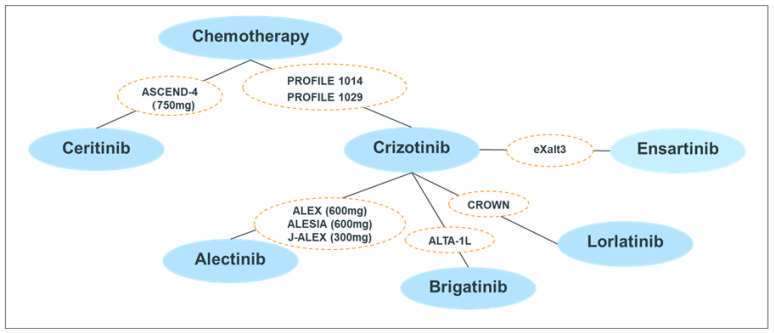
Network plot of included studies.

**Figure 2 jcm-11-02963-f002:**
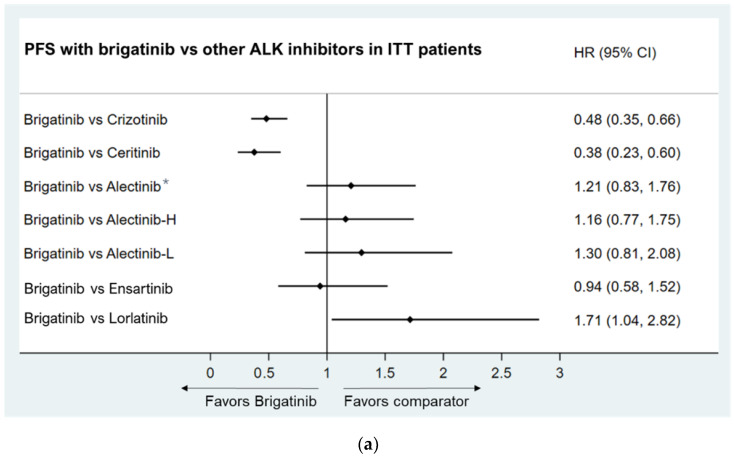
(**a**) Brigatinib vs. other ALK inhibitors in PFS. (**b**) Brigatinib vs. other ALK inhibitors in PFS in Asian patients. (**c**) Brigatinib vs. other ALK inhibitors in PFS in patients with baseline brain metastases. Note: * The relative benefit of alectinib versus crizotinib was demonstrated by pooled results of ALEX, ALESIA, and J-ALEX; Alectinib-H (high-dose alectinib): pooled ALEX and ALESIA study results; Alectinib-L (low-dose alectinib): included J-ALEX study results.

**Figure 3 jcm-11-02963-f003:**
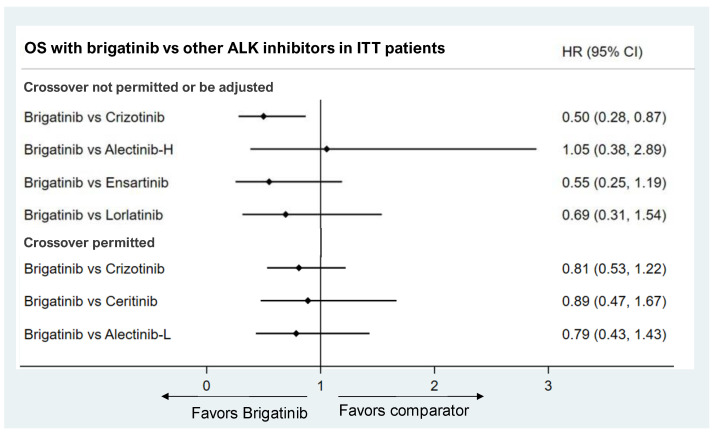
Brigatinib vs. other ALK inhibitors in OS. Note: Alectinib-H (high-dose alectinib): pooled ALEX and ALESIA study results; Alectinib-L (low-dose alectinib): included J-ALEX study results.

**Figure 4 jcm-11-02963-f004:**
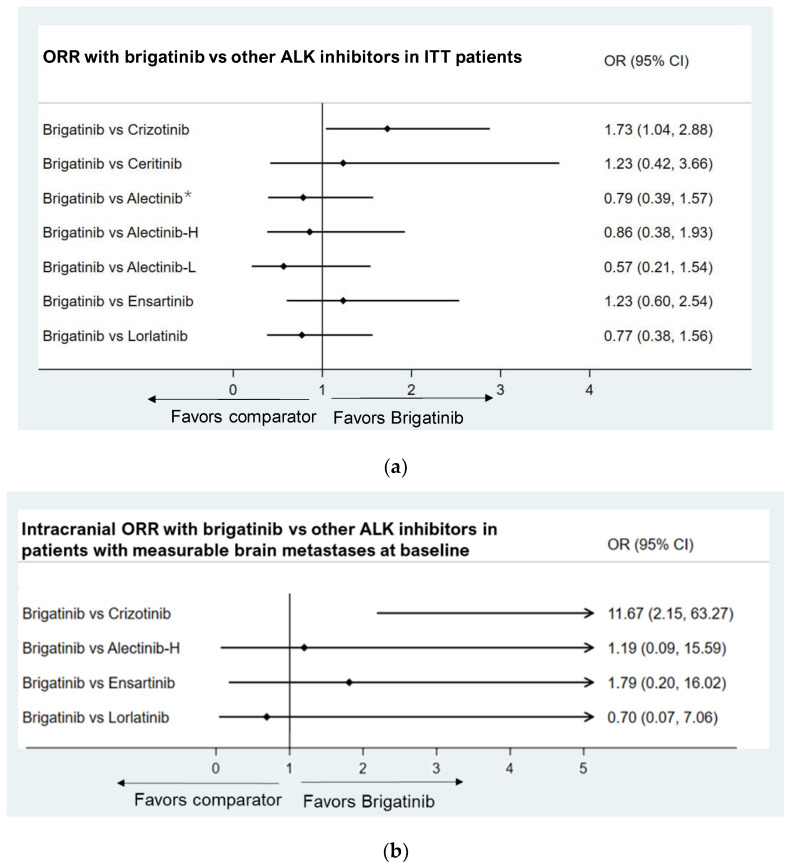
(**a**) Brigatinib vs. other ALK inhibitors in ORR. Note: * The relative benefit of alectinib versus crizotinib was demonstrated by pooled results of ALEX, ALESIA, and J-ALEX; Alectinib-H (high-dose alectinib): pooled ALEX and ALESIA study results; Alectinib-L (low-dose alectinib): included J-ALEX study results. (**b**) Brigatinib vs. other ALK inhibitors in intracranial/CNS ORR in patients with measurable brain metastases at baseline. Note: Alectinib-H (high-dose alectinib): pooled ALEX and ALESIA study results.

**Figure 5 jcm-11-02963-f005:**
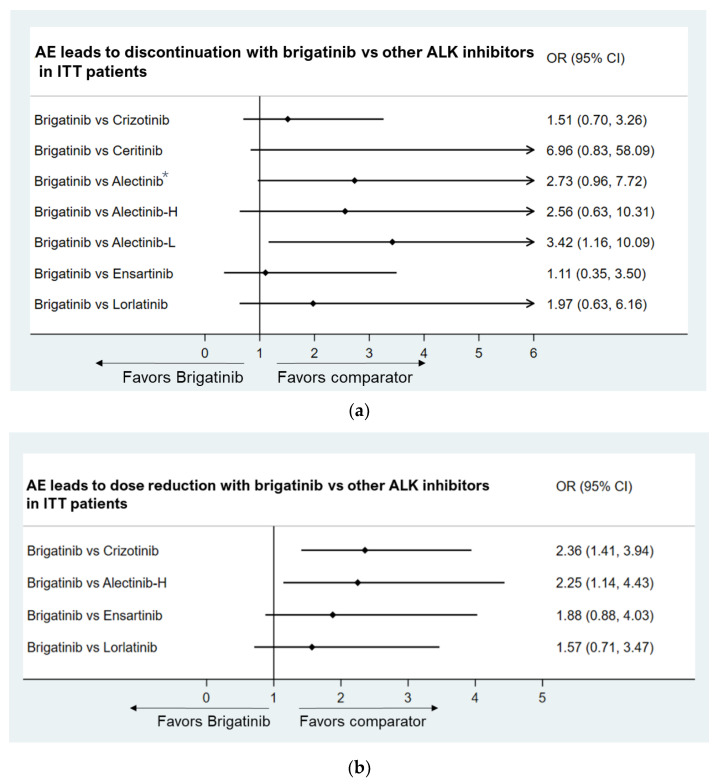
(**a**) Brigatinib vs. other ALK inhibitors in AEs leading to discontinuation. (**b**) Brigatinib vs. other ALK inhibitors in AEs leading to dose reduction. Note: * The relative benefit of alectinib versus crizotinib was demonstrated by pooled results of ALEX, ALESIA, and J-ALEX; Alectinib-H (high-dose alectinib): pooled ALEX and ALESIA study results; Alectinib-L (low-dose alectinib): included J-ALEX study results.

## Data Availability

Not applicable.

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
