# Peer review of "Comparison of Efficacy and Safety of Brigatinib in First-Line Treatments for Patients with Anaplastic Lymphoma Kinase-Positive Non-Small-Cell Lung Cancer: A Systematic Review and Indirect Treatment Comparison"

_jcm, 2022, doi:10.3390/jcm11112963_

Round 1
Reviewer 1 Report
The authors presented a systematic review about brigatinib as first-line treatment in ALK-Positive NSCLC patients.
The paper is clear and well written, but I have some small suggestions.
Abstract: I think that the background parts needs to be revised, since it seems more the aims of the paper and not the description of the state of the art. Please, revise by referring to the first part of the introduction, starting by briefly explain the medical problem and the gap that this review wants to overcome. Background in the abstract represents a very brief summary that the authors could extract from their introduction.
Please, explain all the acronymous that have been used in the abstract (lines 20 ITC, etc.) and then repeat all of them in the main text the first time they appear. Please, revise the main text avoiding the repetition of acronymous explanation twice (line 102 ORR, that has been already explained at line 89-90).
Main text: line 148: I think it should be ITC population.
Lines 361-362: I think the authors could delate the last sentence, since they have explained the conflict of interest of each author.
Author Response
Point1: Abstract: I think that the background parts needs to be revised, since it seems more the aims of the paper and not the description of the state of the art. Please, revise by referring to the first part of the introduction, starting by briefly explain the medical problem and the gap that this review wants to overcome. Background in the abstract represents a very brief summary that the authors could extract from their introduction.
Response: Thank you for your valuable suggestion, we have re-written the background part of the abstract accordingly.
Point2: Please, explain all the acronymous that have been used in the abstract (lines 20 ITC, etc.) and then repeat all of them in the main text the first time they appear. Please, revise the main text avoiding the repetition of acronymous explanation twice (line 102 ORR, that has been already explained at line 89-90).
Response: Thank you for your careful reading and valuable suggestions, we have thoroughly checked the abstract and the main text to see if all the acronymous have been explained and delete the repetition of acronymous explanation.
Point3: Main text: line 148: I think it should be ITC population.
Response: Thank you for your valuable suggestions, ITT refers to the intent-to-treat, and we have added its explanation in the abstract and main text.
Point4: Lines 361-362: I think the authors could delate the last sentence, since they have explained the conflict of interest of each author.
Response: Thank you for your valuable suggestions, we deleted the last sentence accordingly.
Reviewer 2 Report
Thank you for the chance you gave me to read this interesting study entitled “Comparison of efficacy and safety of brigatinib in first-line treatments for patients with anaplastic lymphoma kinase-positive non-small cell lung cancer: A systematic review and indirect treatment comparison” by Yu et al. In this systematic review, the authors present the results from the published randomized controlled trials, focusing on the efficacy and safety of brigatinib compared with other ALK inhibitors for the first-line treatment of patients with ALK-positive non-small cell lung cancer (NSCLC). In this systematic review, 9 randomized controlled trials with 2,484 patients were included. Although this is a very interesting topic and the study is well-written, there are significant issues with this study.
Some of them are:
- Discussion section needs to be re-organized and re-written. More creative presentation of the findings as well as interpretation of the results are necessary. In addition, the results should be removed from the discussion section.
- There are many concerns regarding the novelty of the study, since numerous similar studies have been published the last few years. For instance, few months ago, Ma et al., using 9 randomized controlled trials and 2,484 patients, have presented their meta-analysis, documenting different conclusions.
- The results should be presented without any kind of favoritism.
- Very high similarity rate (42%) based on the Turnitin.
Author Response
Point1: Discussion section needs to be re-organized and re-written. More creative presentation of the findings as well as interpretation of the results are necessary. In addition, the results should be removed from the discussion section.
Response: Thank you for your valuable comments, we have refined the discussion part and more focused on explaining our main findings and the results have been removed accordingly.
Point2: There are many concerns regarding the novelty of the study, since numerous similar studies have been published the last few years. For instance, few months ago, Ma et al., using 9 randomized controlled trials and 2,484 patients, have presented their meta-analysis, documenting different conclusions.
Response: In this work, we have included all the ALK inhibitors and the most updated trials data. We evaluated not only progression free survival, but also overall survival benefit among ALK inhibitors. As for safety, we have learnt from physicians' consultation that AE leading to dose reduction and AE leading to discontinuation which are crucial messages for clinical practice and have further discussed in our analysis. In addition, except for Intent-to-treat population, further analyses have been conducted in patients with baseline brain metastases and Asian patients. We noticed there is a latest article (Ma et al 2021) published in the BMC cancer has similar study design. However, we want to mention that the different issues discussed in our analysis. First, our analysis of alectinib were distinguished between blinded independent review committee vs. investigator assessed results, and treatment cross over. Second, we used the most up-to-date data. Third, we have further discussed the relative efficacy in Asian groups. These results are clinically important given the lack of head-to-head studies. Our study revealed findings that would potentially benefit further future related research and physician in China and other countries. We feel this would be of interest to your audience. As for the conclusions, due to the different methodology, our results mainly revealed the comparative results between brigatinib and other ALK inhibitors, while (Ma,2021) has revealed the efficacy ranking results of ALK inhibitors.
Point3: The results should be presented without any kind of favoritism.
Response: Thank you for your valuable suggestion, we have refined the wording of the whole manuscript in a more objective way.
Point4: Very high similarity rate (42%) based on the Turnitin.
Response: Yes, the methodology and the way of interpreting results of the NMA is similar to some extend among studies but we still improved the writing of our analysis.
Reviewer 3 Report
Although the article is focused on brigatinib and sponsored by Takeda, this is clearly exposed and has been never hidden by the authors. This context has been managed adequately in y opinion. In a clear way, the authors aknowledge that brigatinib is comparable to other new generation competitors and this is fair.
Author Response
Thanks for reading our paper carefully and giving the above positive comments.
Reviewer 4 Report
The authors nicely describe the issue at hand and why their study was necessary. The manuscript is nicely written and concise.
Here are some points that can further improve the manuscript.
Abstract: Too long, can you make it more concise and summarize the main take-home points, especially in the results section? In the introduction part of the abstract, more of a description of the problem at hand is needed. In the methods section, the wording behind "A systemic review was conducted Jan 2010 - Oct 2021, reads that the review was performed over 11 years but instead I believe the authors were trying to say that a review was performed for multiple studies conducted over this time frame (check wording).
Introduction: Lung cancer is a leading cause of cancer-related mortalities worldwide (not only in China).
Methods/Results: Can the authors please include include a more detailed explanation for how confounding variables were eliminated as much as possible between the different RCTs, when comparing the different patients (the 2,484 patients). The schematic in Figure 1 is a little confusing when looking only at the figure, and not reading the text. Can this figure please be modified, perhaps even into a table, to better portray the network of included studies? Each figure comparing Birgatinib and other ALK inhibitors should have a title included in the figure itself (not only in the legend) to better guide the author on what the comparison is. In addition, a column indicating statistical significance (NS or S) would be helpful in accessing the data.
Discussion: As the authors mentioned, one limitation of the study was the use of conference abstracts that may not have included complete information - since these were only abstracts and not solid studies, I would recommend that the authors remove this information from their analysis to avoid any bias/errors in data interpretation.
Author Response
Point1: Abstract: Too long, can you make it more concise and summarize the main take-home points, especially in the results section? In the introduction part of the abstract, more of a description of the problem at hand is needed. In the methods section, the wording behind "A systemic review was conducted Jan 2010 - Oct 2021, reads that the review was performed over 11 years but instead I believe the authors were trying to say that a review was performed for multiple studies conducted over this time frame (check wording).
Response: Thank you for your valuable comments, we have re-written the background part and accordingly and have refined the wording of methods part.
Point2: Introduction: Lung cancer is a leading cause of cancer-related mortalities worldwide (not only in China).
Response: Thank you for your valuable suggestions, we have added epidemiologic statistics of lung cancer worldwide in the beginning of the introduction part.
Point3: Methods/Results: Can the authors please include a more detailed explanation for how confounding variables were eliminated as much as possible between the different RCTs, when comparing the different patients (the 2,484 patients). The schematic in Figure 1 is a little confusing when looking only at the figure, and not reading the text. Can this figure please be modified, perhaps even into a table, to better portray the network of included studies?
Response: Thank you for your valuable suggestions, our analysis have strictly followed the principles of conducting NMA which including establishing inclusion and exclusion criteria before screening studies and only the studies meet the pre-determined criteria can they be included. As for the network plot, it is more to reveal the comparator arm and the common comparators between different comparisons. The detailed information of search strategy, inclusion and exclusion criteria and included studies has been illustrated in the supplementary materials.
Point4: Each figure comparing Birgatinib and other ALK inhibitors should have a title included in the figure itself (not only in the legend) to better guide the author on what the comparison is. In addition, a column indicating statistical significance (NS or S) would be helpful in accessing the data.
Response: Thank you for pointing out this issue in our manuscript. we have redrawn Figure 2a-Figure5b accordingly. It is more common to interpret the ITC results using forest plot as an essential tool. Whether the comparison between A and B is significantly different can be judged by if the horizontal line (the width represents the range of 95% CI) intersects the vertical line in the center of plot (so called “no effect line”).
Point5: Discussion: As the authors mentioned, one limitation of the study was the use of conference abstracts that may not have included complete information - since these were only abstracts and not solid studies, I would recommend that the authors remove this information from their analysis to avoid any bias/errors in data interpretation.
Response: Thank you for your valuable suggestions, we have re-evaluated the conference abstracts we have included, the 2 conference abstracts we included is mainly about revealing the latest trial results, it may cause limited bias to our study results, thus, we have revised the wording of limitation.
Reviewer 5 Report
I found the review : „Comparison of efficacy and safety of brigatinib in first-line 2 treatments for patients with anaplastic lymphoma kinase-posi- 3 tive non-small cell lung cancer: A systematic review and indi- 4 rect treatment comparison” very informative and up to date.
Although I have some small remarks :
- Since there are not only Chinese patients included , please ad some world lunch cancer statistic in introduction section.
- Figure 1-3 and 5 describes Brigatinib on the left and comparator on the right, figures 4 the opposite, that might be a little bit confusing. Unless it has any reasonable explanation please change.
- In the discussion section regarding A, lorlatinib is described as having the higher risk of AE 3-4, however in lines 230-233 it states: ITC results for safety profile suggested that the incidence of grade ≥3 AEs of brigatinib is comparable to ceritinib (OR=0.41, 95% CI: 0.09, 1.91), ensartinib (OR=1.45, 95% CI: 231 0.71, 2.94) and lorlatinib (OR=0.91, 95% CI: 0.43, 1.89), while crizotinib and alectinib pre- 232 sented a lower incidence rate of grade ≥3 AEs (OR=1.99, 95% CI: 1.17, 3.41 for crizotinib, 233 OR=3.72, 95% CI: 1.77, 7.83 for alectinib). More over the only AE regarding this drug is described in detail ( mood changes). This is rather subjective , not describing typical AE’s of other TKI.
- The article is well written, perfect English
Author Response
Point1: Since there are not only Chinese patients included, please add some world lunch cancer statistic in introduction section.
Response: Thank you for your valuable suggestions, we have added epidemiologic statistics of lung cancer worldwide in the beginning of the introduction part.
Point2: Figure 1-3 and 5 describes Brigatinib on the left and comparator on the right, figures 4 the opposite, that might be a little bit confusing. Unless it has any reasonable explanation please change.
Response: Thank you for your valuable suggestions, we’d like to clarify the arrows orientation in evaluating different efficacy endpoint.
1) Objective response rate (ORR) is a positive indicator; the effect measurement of ORR is odds ratio (OR). When compared brigatinib with other ALK inhibitors, the higher odds ratio indicates that brigatinib has higher objective response rate, hence, the “favor brigatinib” arrow is presented on the right side of forest plot (OR>1). (as Figure 4)
2) On the opposite, the effect measurement of PFS and OS is Hazard ratio (HR). When compared brigatinib to other ALK inhibitors, the higher HR indicates that birgatinib has higher progression or death risk, hence, the “favor brigatinib” arrow is presented on the left side of forest plot (HR<1) (as Figure1-3).
Point3: In the discussion section regarding A, lorlatinib is described as having the higher risk of AE 3-4, however in lines 230-233 it states: ITC results for safety profile suggested that the incidence of grade ≥3 AEs of brigatinib is comparable to ceritinib (OR=0.41, 95% CI: 0.09, 1.91), ensartinib (OR=1.45, 95% CI: 231 0.71, 2.94) and lorlatinib (OR=0.91, 95% CI: 0.43, 1.89), while crizotinib and alectinib pre- 232 sented a lower incidence rate of grade ≥3 AEs (OR=1.99, 95% CI: 1.17, 3.41 for crizotinib, 233 OR=3.72, 95% CI: 1.77, 7.83 for alectinib). More over the only AE regarding this drug is described in detail ( mood changes). This is rather subjective, not describing typical AE’s of other TKI.
Response: Thank you for your valuable suggestions, we have removed the AE of lorlatinib.
Point4:The article is well written, perfect English
Response: Thanks for reading our paper carefully and giving the positive comments.